# Long-term live imaging, cell identification and cell tracking in regenerating crustacean legs

**Çağrı Çevrim[1,2†], Béryl Laplace-Builhé[1,2†], Ko Sugawara[1,2,3†], Maria Lorenza Rusciano[1,2], Nicolas Labert[1,4], Jacques Brocard[5,6], Alba Almazán[1,2], Michalis Averof[1,2]***

[1]Institut de Génomique Fonctionnelle de Lyon (IGFL), École Normale Supérieure de Lyon, Lyon, France; [2]Centre National de la Recherche Scientifique (CNRS), Paris, France; [3]Laboratory for Developmental Dynamics, RIKEN Center for Biosystems Dynamics Research (BDR), Kobe, Japan; [4]Université Claude Bernard Lyon 1, Lyon, France; [5]PLATIM Imaging Facility, SFR Biosciences, École Normale Supérieure de Lyon, Lyon, France; [6]Institut National de la Santé et de la Recherche Médicale (INSERM), Paris, France

## eLife Assessment

This study presents a **valuable** technical advance in the long-term live imaging of limb regeneration at cellular resolution in *Parhyale hawaiensis*. The authors develop and carefully validate a method to continuously image entire regenerating legs over several days while minimizing photodamage and optimizing conditions for robust cell tracking, together with post-hoc in situ identification of cell types. The data are **convincing**, the methodology is rigorous and clearly documented, and the results will be of interest to researchers in regeneration biology, developmental biology, and advanced live imaging and cell tracking software development.
[Editors' note: this paper was reviewed by Review Commons.]

**\*For correspondence:**
michalis.averof@ens-lyon.fr

[†]These authors contributed equally to this work

**Abstract** High-resolution live imaging of regeneration presents unique challenges due to the nature of the specimens (large mobile animals), the duration of the process (spanning days or weeks), and the fact that cellular resolution must be achieved without damage caused by lengthy exposures to light. Building on previous work that allowed us to image different parts of the process of leg regeneration in the crustacean *Parhyale hawaiensis*, we present here a method for live imaging that captures the entire process of leg regeneration, spanning up to 10 days, at cellular resolution. Our method includes (1) mounting and long-term live imaging of regenerating legs under conditions that yield high spatial and temporal resolution but minimise photodamage, (2) fixing and in situ staining of the regenerated legs that were imaged, to identify cell fates, and (3) computer-assisted cell tracking to determine the cell lineages and progenitors of identified cells. The method is optimised to limit light exposure while maximising tracking efficiency. Combined with appropriate cell-type-specific markers, this method may allow the description of cell lineages for every regenerated cell type in the limb.

## Introduction

In contrast to imaging embryonic development, live imaging of regeneration presents unique challenges that have so far been difficult to overcome. First, unlike developing embryos which can be easily immobilised, regenerating juvenile or adult animals are mobile and cannot easily be fixed under a microscope over long periods of time. These animals need to move to capture food and, in many cases, to breathe. Anaesthesia is possible for short periods, but lethal if applied over many hours or days.

Second, imaging embryonic development has been particularly successful in animals with rapid development. Developmental processes with a duration of hours or a few days, such as the embryonic development of *C. elegans*, *Drosophila*, and zebrafish, or the process of somitogenesis in mice (*Keller et al., 2008*; *Masamizu et al., 2006*; *Sulston et al., 1983*; *Tomer et al., 2012*), can be followed with high temporal resolution. Processes with a longer duration, however, present greater challenges due to the damaging effects that prolonged exposure to light has on living tissues (*Icha et al., 2017*). Long-term live imaging also requires robust microscope-camera setups that can maintain image quality and record images over long periods. Regeneration typically takes place on a timescale of several days or weeks.

Third, the larger size of juvenile or adult organisms can impact live imaging in a number of ways, for example, by constraining how large specimens can be mounted on a microscope (or, indeed, how a microscope might be mounted on the animal, see *Aharoni et al., 2019*), or by exceeding a microscope's field of view or imaging depth.

In spite of these difficulties, live imaging of regeneration – under anaesthesia – has been developed in zebrafish and mice, allowing for successful imaging of cell dynamics during repair or regeneration in different organs (*Chioccioli et al., 2024*; *Cox et al., 2018*; *Gurevich et al., 2016*; *Kamimoto et al., 2020*; *Mateus et al., 2012*; *Park et al., 2017*; *Pineda et al., 2015*; *Rompolas et al., 2012*; *Webster et al., 2016*). The duration of continuous time-lapse recordings, in these cases, has been constrained by poor survival under long-term anaesthesia and is typically limited to less than one day. An alternative approach has been to immobilise a regenerating animal for the time required to acquire an image stack and then to release it, until the next imaging time point when one re-captures the animal and re-visits the same body part. This approach has allowed imaging experiments to run over much longer time periods (several days, even weeks), but with much lower temporal resolution, typically 12 or 24 h between successive time points (*Chen et al., 2016*; *Chioccioli et al., 2024*; *Cox et al., 2018*; *Currie et al., 2016*; *De Simone et al., 2021*; *Mateus et al., 2012*; *Pineda et al., 2015*; *Rompolas et al., 2012*).

A major challenge in the latter approach is identifying the same cells at different time points, especially after cell divisions and movements. This can be made easier by sparse labelling of cells, or mosaic labelling of cell clones, for example using *Brainbow*-like methods (*Barbosa et al., 2015*; *Chen et al., 2016*; *Currie et al., 2016*; *Ritsma et al., 2014*).

In recent years, the crustacean *Parhyale hawaiensis* has emerged as a promising experimental system for studying regeneration, in which transgenic tools can be combined with live imaging (*Alwes et al., 2016*; *Grillo et al., 2016*; *Konstantinides and Averof, 2014*; *Paris et al., 2022*). For live imaging, a major advantage of *Parhyale* is our ability to immobilise its regenerating legs over long periods without needing to keep the animals under anaesthesia. This is achieved by fixing the chitinous exoskeleton that surrounds the leg onto the microscope coverslip using surgical glue (*Alwes et al., 2016*; *Konstantinides and Averof, 2014*). Thus, *Parhyale*'s cuticle – which is both sturdy and transparent – serves as a straitjacket to immobilise the leg, and as a window through which we can visualise regeneration. Our first systematic study using this approach presented continuous live imaging over periods of 2–3 days, capturing key events of leg regeneration such as wound closure, cell proliferation and morphogenesis of regenerating legs at single-cell resolution (*Alwes et al., 2016*). Here, we extend this work by developing a method for imaging the entire course of leg regeneration, optimised to reduce photodamage and to improve cell tracking. We also develop a method of in situ staining of gene expression in cuticularised adult legs, which we combine with live imaging to determine the fate of tracked cells.

Our objective is to image legs during the entire time course of leg regeneration, with sufficient spatial and temporal resolution to support the tracking of individual cells in 3D. We also want to determine the identities and molecular signatures of the tracked cells. In the long term, this will enable us

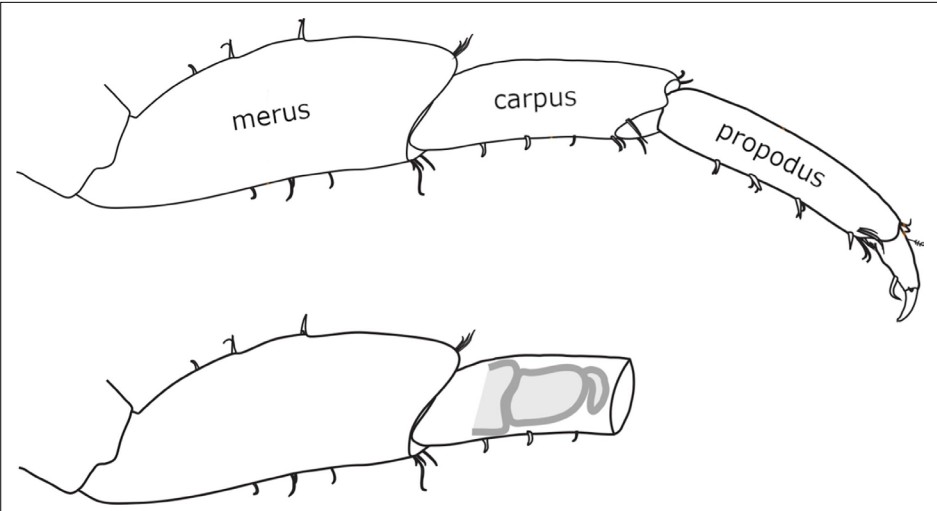

**Figure 1.** Morphology and regeneration of *Parhyale* legs. Illustration of the distal-most podomeres of an intact *Parhyale* T4 or T5 leg, including the merus, carpus, and propodus (top), and of a leg amputated at the distal part of the carpus (bottom). In the amputated leg stump, the site where regeneration takes place is illustrated with a cartoon of the regenerating leg. Our live regeneration experiments focus on that region.

to identify progenitors for each cell type in the leg, to trace their complete genealogies, and to characterise the molecular and cell state transitions associated with leg regeneration.

Leg regeneration takes place over approximately 1 week in adult *Parhyale* (3–10 days, depending on the age of the individual, the injury sustained, and other unknown factors) (*Alwes et al., 2016*; *Konstantinides and Averof, 2014*; *Sinigaglia et al., 2022*). Our previous studies showed that imaging at 20-min intervals is sufficient to capture cell divisions and to track cells reliably over time (*Alwes et al., 2016*). Thus, a complete recording of a regenerating leg would require capturing at least 360 image stacks, at 20-min intervals, over a period of 5 days. A major challenge we must address in such long-term imaging experiments is balancing the need for high spatial and temporal resolution with the need to minimise photodamage. Increasing spatial or temporal resolution (e.g. by using higher numerical aperture objectives, finer sampling in *x*, *y* or *z*, or shorter imaging intervals) improves our ability to resolve and to track cells in a crowded cellular environment, but also leads to increased light exposure and resulting tissue damage.

Among the various imaging modalities available for live imaging (*Schroeder, 2008*), we have used confocal microscopy. Light-sheet microscopy would have been preferable for minimising light exposure (*Stelzer et al., 2021*), but the currently accessible light-sheet designs are incompatible with mounting and imaging through a glass coverslip, which we need in order to immobilise our specimens. Confocal microscopy is now widely established and commercial systems are stable enough to support automated 3D image acquisition over very long time courses.

## Results and discussion

### Imaging the entire course of leg regeneration at cellular resolution

*Parhyale* can regenerate their limbs independently of the site of amputation. Regeneration occurs in the distal part of the limb stump, within the exoskeleton of the amputated leg (*Figures 1 and 2*; *Alwes et al., 2016*; *Konstantinides and Averof, 2014*). We have found that by amputating the T4 or T5 legs of mid-sized adults at the distal part of the carpus, we can capture the entire regenerating tissue within a single field of view using a 20× objective (Zeiss Plan-Apochromat 20×/0.8) (see *Figure 2*, *Videos 1 and 2*).

Live imaging is performed on transgenic animals expressing a histone-bound fluorescent protein expressed under a heat-shock promoter (*Alwes et al., 2016*; *Pavlopoulos et al., 2009*; *Wolff et al., 2018*). To minimise photodamage, we image at long wavelengths, using a construct expressing H2B-mRFPruby (*Wolff et al., 2018*). To induce transgene expression, a heat shock (45 min at 37°C) is

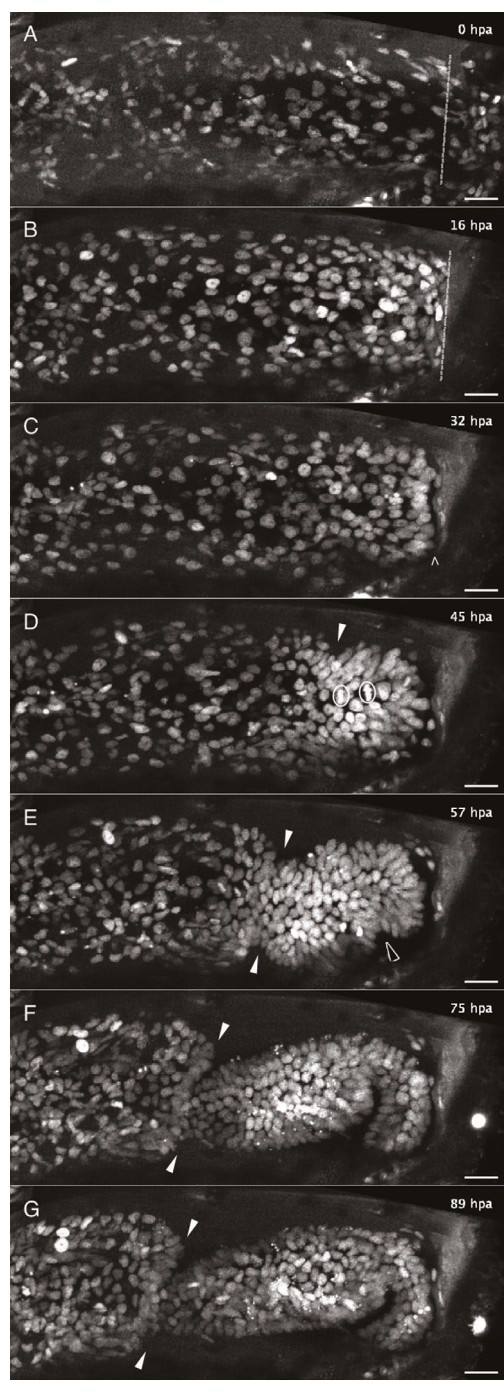

**Figure 2.** Live imaging capturing the phases of leg regeneration in *Parhyale*. Phases of leg regeneration were observed by live imaging of nuclei labelled with H2B-mRFPruby (images from dataset li48-t5; *Video 1*). Proximal parts of the leg are to the left and the amputation site is on the right of each panel. (**A**) T5 leg imaged shortly after amputation, showing haemocytes adhering to the wound site (on the right of dashed line). (**B**) At 16 hpa, haemocytes have produced a melanised scab at the wound; epithelial cells are migrating below the scab. (**C**) At 32 hpa, the

*Figure 2 continued on next page*

*Figure 2 continued*

leg tissues have become detached from the scab (open arrow). (**D**) At 45 hpa, the new carpus–propodus boundary becomes visible (white arrow); dividing cells can be seen at the distal part of the leg stump (mitotic figures marked by circles). (**E**) At 57 hpa, the propodus–dactylus boundary becomes visible (black arrow); the carpus and propodus are well separated (white arrows). (**F, G**) In later stages, tissues in more proximal parts of the leg (to the left of the white arrows) retract, making space for the growing regenerating leg. hpa: hours post amputation. Scale bars, 20 μm.

The online version of this article includes the following figure supplement(s) for figure 2:

**Figure supplement 1.** Testing the effects of scanning speed and averaging on image quality.

**Figure supplement 2.** Apoptosis in legs that have not been subjected to live imaging.

applied, typically 12–18 h before amputation, yielding high levels of nuclear fluorescence which can persist for many days in non-dividing cells. Heat shocks may be repeated during the course of imaging, using the microscope's temperature-controlled stage, to maintain adequate levels of fluorescence in dividing cells. The temperature change causes a slight displacement in the acquired images, which is corrected post acquisition (see Methods).

To minimise light-induced damage, we select transgenic individuals that show the highest levels of nuclear fluorescence and set the power of the excitation laser to the lowest setting that yields acceptable image quality on our most sensitive GaAsP detector. Previous experiments had shown that imaging with a pixel size of 0.31 × 0.31 μm, a *z* step of 2.48 μm, and a time interval of 20 min is sufficient for cell tracking (*Alwes et al., 2016*), both manually and using the Elephant tracking software (*Sugawara et al., 2022*). To improve image quality while keeping light exposure in check, we have tested different settings for scanning speed and image averaging and determined that a scanning speed of 2.06 μs per pixel with an averaging of two images (or 1.03 μs per pixel with an averaging of four images) are the fastest settings that generate images of sufficient quality for tracking (*Figure 2—figure supplement 1*). Most animals imaged under these settings regenerate their legs within 5–10 days from amputation (*Figure 3*), unless they have been amputated shortly before molting. In the latter case, the wound is closed and the animals molt before regenerating. The imaging settings are described in more detail in the Methods.

By continuously imaging the entire process of leg regeneration, we are now able to corroborate and extend our previous descriptions of leg regeneration, which were reconstructed from snapshots and partial recordings (*Alwes et al., 2016*; *Konstantinides and Averof, 2014*). Within 24 hours post amputation (hpa), a melanised scab forms at the site of injury (*Figure 2A, B*). This is accompanied by stretching and migration of epithelial cells to cover the wounded surface, under the scab (10–30 hpa in *Videos 1 and 2*). There is little (if any) cell division during these early stages (*Figure 4*). Between 1 and 3.5 days post amputation, the epithelium detaches from the scab (*Figure 2C*; 31 hpa in *Video 1*, 65 hpa in *Video 2*). The timing of detachment varies considerably (*Figure 3*), but we find that it often coincides with the onset of cell proliferation, cell–cell rearrangements and apoptosis in the cells that lie within ~100 µm from the scab. More proximally located tissues detach from the cuticle and retract towards the body, making space for the growth of regenerating tissues in the distal part of the stump (40–85 hpa in *Video 1*; 80–120 hpa in *Video 2*). The boundaries of the leg podomeres typically appear within a day from epithelial detachment (*Figure 2D, E*; 45–60 hpa in *Video 1*; 80–95 hpa in *Video 2*). In some recordings, the tissues making up the podomeres can be seen pulsating with a periodicity of 6–7 h (see 105–145 hpa in *Video 2*). This pulsating movement had not been observed previously, and its significance is unclear. Later, the rates of cell proliferation and movement gradually decline and the nuclear positions within the tissue become stabilised (*Figure 2G*). Based on transcriptome profiling data, we think that many cells differentiate during this phase (*Sinigaglia et al., 2022*). Once regeneration is completed, the newly regenerated leg is trapped inside the exoskeleton of the amputated leg stump. It is released from the old exoskeleton and becomes functional during the animal's next molt. When the animal molts, the regenerated leg is fully formed (*Almazán et al., 2022*). Molting allows the animal to detach from the coverslip and therefore marks the end of our imaging.

The timing of these events varies from leg to leg, but their order is consistent in all our recordings (summarised in *Figure 3*). Our fastest recordings show the leg of a young adult regenerating in less than 3 days at 29°C (li51-t4 and li51-t5). In contrast, the longest ones take ~10 days to complete regeneration (li16-t5 and li34-t4, imaged at 26–27°C). We find that the overall speed of leg regeneration is determined largely by variation in the speed of the early (wound closure) phase of regeneration, and to a lesser extent by variation in later phases when leg morphogenesis takes place (*Figure 3—figure supplement 1A, B*).

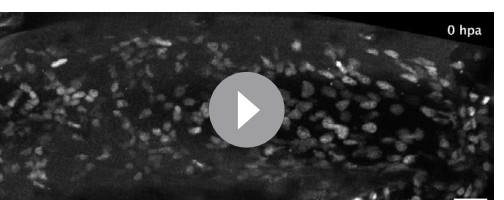

**Video 1.** Time-lapse recording of regeneration in a *Parhyale* T5 leg (dataset li48-t5). Live imaging of nuclei labelled with H2B-mRFPruby (maximum projection of *z* slices 3–10). Proximal parts of the leg are to the left and the amputation site is at the right of the frame. For annotations of different features, please refer to *Figure 2*. Shortly after leg amputation (0 hpa), haemocytes adhere to the wound. By 16 hpa, the wound has melanised. Up to ~32 hpa epithelial cells can be seen migrating and accumulating at the wound, below the melanised scab (*Figure 2A, B*). Around 31 hpa, the leg tissues become detached from the scab (*Figure 2C*). At 43 hpa, the carpus–propodus boundary first becomes visible, and thereafter many cells can be observed dividing at the distal part of the leg stump (*Figure 2D*). At 56 hpa, the propodus–dactylus boundary first becomes visible (*Figure 2E*). At later stages, tissues in more proximal parts of the leg retract, making space for the regenerating leg to grow (*Figure 2F, G*). After ~90 hpa cell proliferation, there is less cell proliferation and cell movements, and the nuclear positions within the tissue become fixed. Scale bars, 20 µm.

https://elifesciences.org/articles/107534/figures#video1

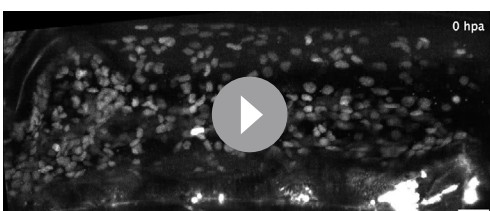

**Video 2.** Time-lapse recording of regeneration in a *Parhyale* T5 leg (dataset li36-t5). Live imaging of nuclei labelled with H2B-mRFPruby (maximum projection of *z* slices 3–15). Proximal parts of the leg are to the left and the amputation site is at the right of the frame. The sequence of events is similar to that described in *Video 1*, but the progression is slower: epithelial migration towards the wound is observed up to 40 hpa, tissues detach from the scab at 65 hpa, and the carpus–propodus and propodus–dactylus boundaries first become visible at 78 and 91 hpa. The tissues making up the carpus and propodus can be seen pulsating from 105 to 145 hpa. Scale bars, 20 µm.

https://elifesciences.org/articles/107534/figures#video2

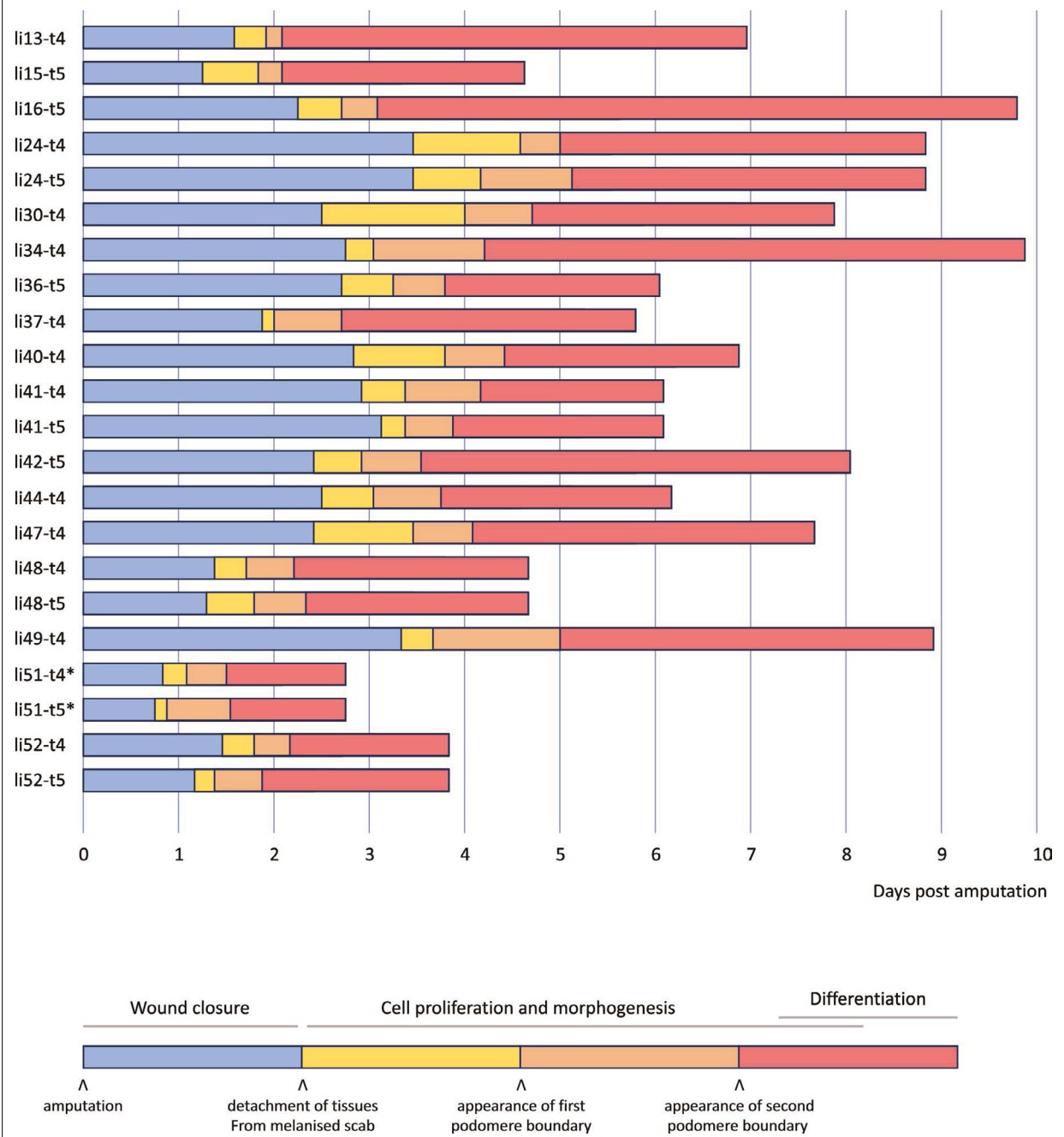

**Figure 3.** Overview of 22 live recordings of *Parhyale* leg regeneration. For each recording, we indicate the span of the early phase of wound closure and proliferative quiescence prior to epithelial detachment from the scab (in blue), the early phases of cell proliferation and morphogenesis, up to the time when the first and second podomere boundaries become visible (in yellow and orange, respectively), and the late phases of regeneration, when the leg takes its final shape, cell proliferation gradually dies down and cells differentiate (in red). The total duration of this process varies from 3 to 10 days. Note that the fastest instances of regeneration occurred in young individuals (li51 and li52); the fastest, li51, was imaged at 29°C (marked by an asterisk). Datasets li-13 and li-16 were recorded until the molt; the other recordings were stopped before molting. Further details on each recording are given in Supplementary Data 1 available at https://doi.org/10.5281/zenodo.15181497.

The online version of this article includes the following figure supplement(s) for figure 3:

**Figure supplement 1.** Variation in the duration of early and late phases of regeneration in relation to the overall speed of leg regeneration.

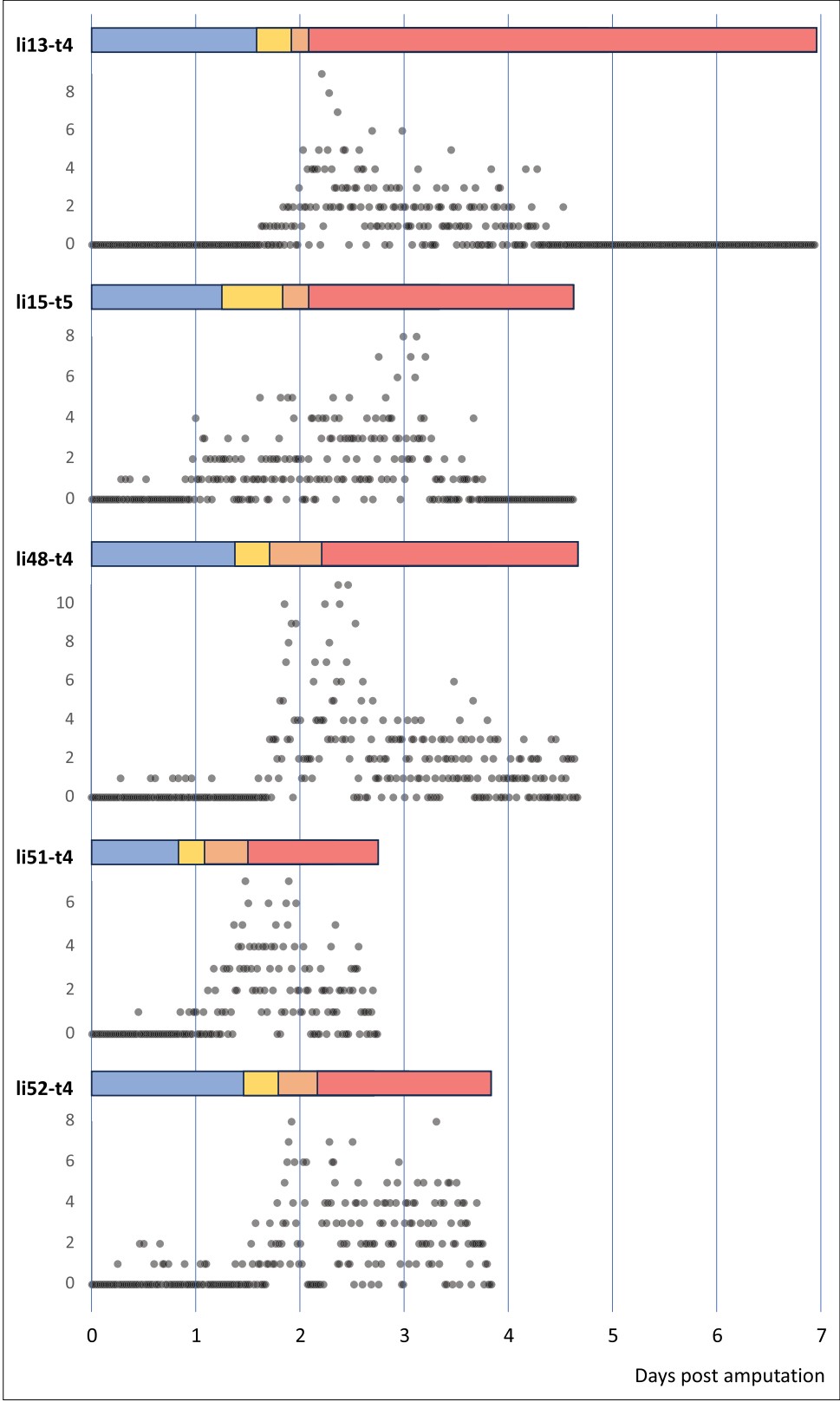

**Figure 4.** Temporal pattern of cell divisions in regenerating *Parhyale* legs. The number of cell divisions detected per time point is shown for five recordings of regenerating legs. The divisions were extracted from tracking data (for li13-t4) or detected using a semi-automated approach (see Methods). The detection of divisions is not exhaustive.

There is no clear relationship between the *relative* duration of each phase and the speed of regeneration (*Figure 3—figure supplement 1 A',B'*).

We note that pairs of adjacent T4 and T5 legs that were amputated simultaneously (in datasets li24, li41, li48, li51, and li5) appear to regenerate at a similar pace, suggesting variation in the speed of regeneration is influenced by systemic factors, such as the age, health or physiology of the individual.

During the course of regeneration, some cells undergo apoptosis (reported in *Alwes et al., 2016*). Using the H2B-mRFPruby marker, apoptotic cells appear as bright pyknotic nuclei that break up and become engulfed by circulating phagocytes (see bright specks in *Figure 2F*). While some cell death might be caused by photodamage, apoptosis can also be observed in regenerating legs that have not been subjected to live imaging (*Figure 2—figure supplement 2*).

## Computer-assisted cell tracking during regeneration

Continuous live imaging provides a unique opportunity to track cells and determine their behaviours, lineages and fates during the course of regeneration. We use the Mastodon cell tracking platform (https://github.com/mastodon-sc/mastodon; *Tinevez et al., 2025*) to perform cell tracking on regenerating *Parhyale* legs. We previously presented Elephant, a Mastodon plug-in that enables semi-automated cell tracking (https://github.com/elephant-track; *Sugawara et al., 2022*). Elephant uses deep learning on sparse annotations provided by the user to identify and track nuclei across an entire image dataset, in 3D. The computationally predicted tracks must then be proofread and validated by the user.

The regenerating legs of *Parhyale* make challenging datasets for cell tracking, due to large variations in signal intensity and poor image quality in the deeper tissue layers. This is exacerbated by the need to limit the tissues' exposure to light, which constrains signal intensity, spatial and temporal resolution (see above). Moreover, the datasets capturing the entire course of regeneration are large, typically consisting of image stacks of 15–30 *z* slices collected over 300–700 time points (i.e. ~10,000 images per dataset), which makes both manual tracking and proofreading of computer-generated tracks very laborious.

The initial implementation of Elephant generated tracking predictions over the entire image stack. We have now added a 'backtracking' function (see Methods), which allows users to focus on tracking cells of particular interest. With this function, users are able to select individual cells and obtain predictions on their progenitors, moving up towards earlier points in their cell lineage. This is particularly useful when specific cells of interest can be identified based on molecular or other markers (see below).

Using this mode of semi-automated cell tracking, we find that most cells in the upper slices of our image stacks (top 30 µm) can be tracked with a high degree of confidence. A smaller proportion of cells is trackable in deeper layers (*Figure 5*).

## Optimising trade-offs between imaging resolution, image quality, and light exposure, based on cell tracking performance

Having identified settings that allow us to image the entire course of leg regeneration in *Parhyale*, we asked whether imaging could be improved further, to facilitate cell tracking, without exposing the legs to additional light. We approached this question by exploring the trade-offs between spatial resolution, temporal resolution and image quality, keeping in mind that the crowding of nuclei (in relation to *z* resolution) and the speed of mitosis (in relation to temporal resolution) are likely to limit tracking accuracy.

We started by recording leg regeneration at higher than usual spatial and temporal resolutions (0.31 × 0.31 µm pixel size, 1.24 µm *z* step, 10-min time intervals, four image replicates) and tracking all the nuclei in this volume to generate ground truth track annotations. We then simulated different imaging conditions by subsampling this recording to generate five datasets that differ in spatial resolution, temporal resolution and/or image quality: (#1) a dataset corresponding to our standard imaging settings, described earlier; (#2) a dataset with improved *z* resolution, but lower image quality; (#3) a dataset with improved temporal resolution, but lower image quality; (#4) a dataset with improved *z* and temporal resolutions, but lower resolution in *x* and *y*; and (#5) a dataset with improved *z* and temporal resolutions, but lower image quality (summarised in *Table 1*). To assess the trade-offs between these parameters, we made sure that each of these five datasets would require precisely

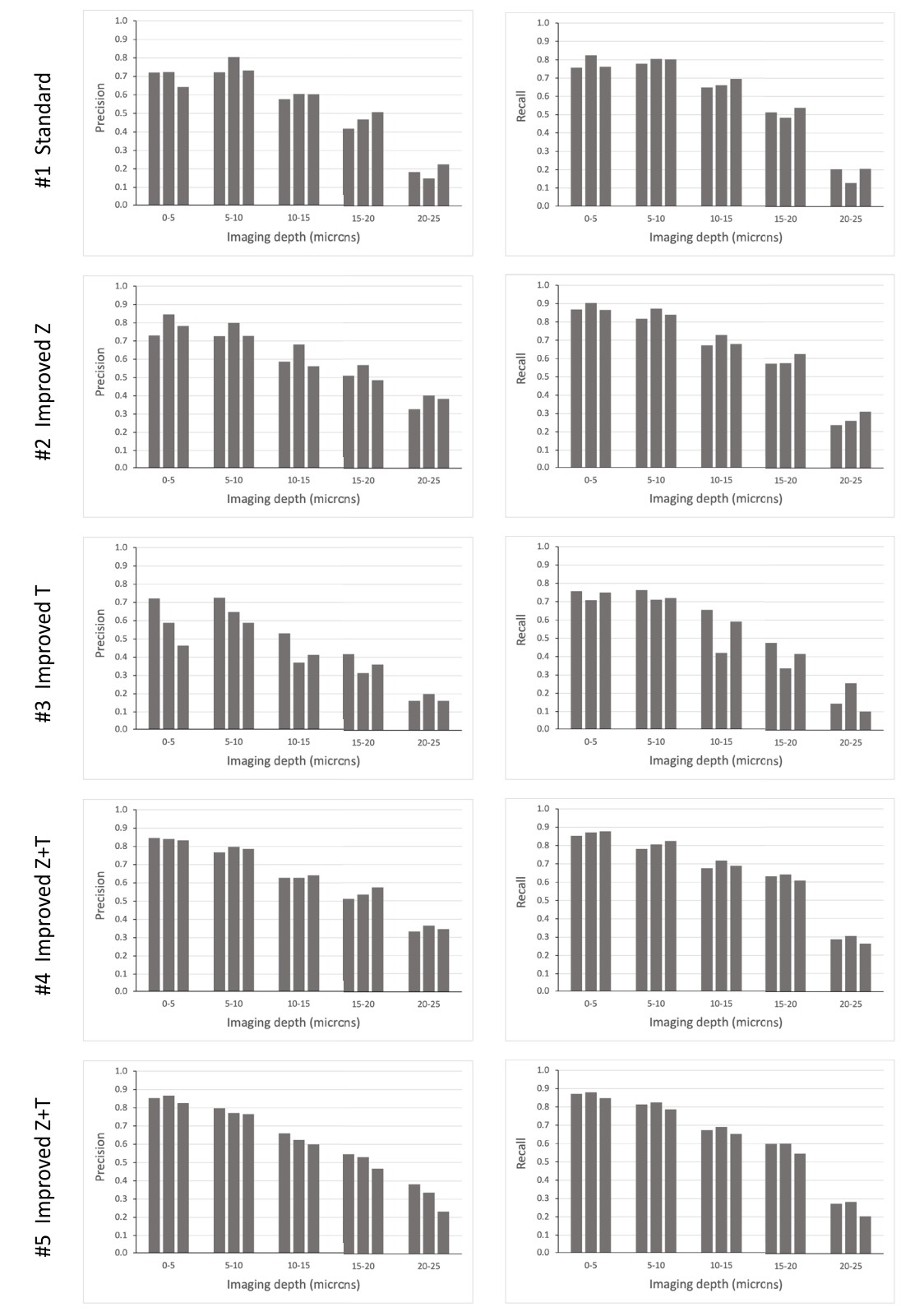

**Figure 5.** Tracking efficiency in relation to imaging depth. Elephant's performance in detecting nuclei at different depths was assessed in the datasets presented in *Table 1A*. Each column represents an independent replicate. Precision and recall of detection (see Methods) were scored at *z* intervals of 5 μm. The data for this figure are provided in Supplementary Data 8 at https://doi.org/10.5281/zenodo.15181497.

**Table 1.** Evaluating image acquisition settings and post-processing settings based on cell tracking performance.

(A) Elephant's tracking performance was evaluated on the same image dataset rendered at different resolutions and image quality (datasets #1–5), as described in the Methods. To measure linking performance independently of detection, the linking metrics were determined for tracking performed on the spots of the ground truth data (TRA on GT). Dataset #1 corresponds to our standard image acquisition settings; improvements in the z and t resolutions are indicated in blue; lowering the xy resolution or the number of averaged replicate images are indicated in red. The best scores obtained for each metric are indicated in purple. To account for variation in the training, three detection and three linking models were trained and evaluated independently on each dataset. The table shows the averaged scores of the three iterations. (B) Elephant's tracking performance was evaluated on dataset #1 with and without post-processing by denoising or deconvolution (see Methods). The metrics are the same as above. Image denoising did not significantly improve cell tracking, and deconvolution (under the setting tested) gave worse results.

| A. Image acquisition | Image settings | | | | CTC scores | | Detection metrics | | Linking metrics | | Proofreading time/effort |
|---|---|---|---|---|---|---|---|---|---|---|---|
| | xy (µm) | z (µm) | t (min) | Averaging | DET | TRA | Precision | Recall | Precision | Recall | |
| Ground truth (4E) | 0.31 × 0.31 | 1.24 | 10 | 4 | 0.71 | 0.70 | 0.68 | 0.74 | 0.99 | 0.99 | 47,588 |
| #1 Standard | 0.31 × 0.31 | 2.48 | 20 | 4 | 0.62 | 0.61 | 0.61 | 0.66 | 0.97 | 0.95 | 31,803 |
| #2 Improved Z | 0.31 × 0.31 | 1.24 | 20 | 2 | 0.68 | 0.67 | 0.66 | 0.72 | 0.97 | 0.95 | 26,426 |
| #3 Improved T | 0.31 × 0.31 | 2.48 | 10 | 2 | 0.58 | 0.57 | 0.53 | 0.64 | 0.99 | 0.99 | 67,783 |
| #4 Improved Z+T | 0.62 × 0.62 | 1.24 | 10 | 4 | 0.68 | 0.67 | 0.68 | 0.71 | 0.99 | 0.99 | 52,253 |
| #5 Improved Z+T | 0.31 × 0.31 | 1.24 | 10 | 1 | 0.67 | 0.66 | 0.68 | 0.70 | 0.99 | 0.99 | 54,280 |

| B. Post-processing | Image settings | | CTC scores | | Detection metrics | | Linking metrics | | Proofreading time/effort |
|---|---|---|---|---|---|---|---|---|---|
| | xyz (µm) | Post-processing | DET | TRA | Precision | Recall | Precision | Recall | |
| Ground truth (4E) | 0.31 × 0.31 × 1.24 | None | 0.71 | 0.70 | 0.68 | 0.74 | 0.99 | 0.99 | 47,588 |
| #1 Standard | 0.31 × 0.31 × 2.48 | None | 0.62 | 0.61 | 0.61 | 0.66 | 0.97 | 0.95 | 31,803 |
| #1 +Denoising | 0.31 × 0.31 × 2.48 | Denoising | 0.62 | 0.61 | 0.64 | 0.65 | 0.97 | 0.95 | 31,838 |
| #1 +Deconvolution | 0.31 × 0.31 × 2.48 | Deconvolution | 0.49 | 0.48 | 0.47 | 0.55 | 0.97 | 0.95 | 42,386 |

the same amount of light exposure. We then evaluated and compared the performance of Elephant across these datasets, in relation to the ground truth, using different metrics, including overall cell detection (DET) and tracking (TRA) scores, measurements of precision and recall rates for the detection and linking of spots across time points, and an estimate of the time or effort required to proofread the resulting cell tracks (see Methods).

Our results, summarised in *Table 1A*, show that the detection of nuclei can be enhanced by doubling the z resolution at the expense of xy resolution and image quality (compare detection metrics in #1 versus #2, #4, and #5). This improvement is particularly evident in the deeper layers of the imaging stacks, which are usually the most challenging to track (*Figure 5*). Tracking of nuclei across time points (linking) can be enhanced by doubling the temporal resolution at the expense of xy resolution and image quality (compare linking metrics in #1 versus #3, #4, and #5). Taking both detection and linking into account, it appears that the best tracking performance is achieved by improving both z and t resolution, at the expense of xy resolution or image quality (conditions #4 and #5). However, doubling the temporal resolution requires much more effort for proofreading (see last column in *Table 1A*), because the datasets contain many more time points.

We conclude that improving the z resolution (condition #2) should be the preferred option when taking into account the efforts required for proofreading. Improving both z and t resolution (condition #4 or #5) will be valuable for tracking more rapid changes, or when users seek the best tracking performance from Elephant, without proofreading (e.g. automated tracking).

We also tested whether image deconvolution or denoising using state-of-the-art algorithms could improve cell detection and tracking. We find that, under the settings tested, deconvolution and denoising did not help to improve cell tracking performance (*Table 1B*).

## Combining live imaging with in situ staining to determine the fate of tracked cells

Nuclear-localised fluorescence is convenient for cell tracking but does not help to identify specific cell types, as cell shapes and cell-to-cell contacts remain invisible. During the last years, we have tried to develop cell-type-specific fluorescent markers for live imaging using different approaches (CRISPR knock-ins, gene trapping, promoter reporters), but we have not yet succeeded in developing such tools. To overcome this limitation, we have turned to in situ staining of molecular markers. Our method involves live imaging, fixation and post hoc staining of the regenerated legs, and manual image registration, allowing us to associate the stained cells with specific nuclei tracked in the live recordings.

Initially, we developed this approach using immunostaining with antibodies for the *Parhyale* Repo protein (identifying putative glial cells, *Almazán et al., 2022*), a cross-reactive antibody raised against Prospero (*Perry et al., 2016*) and an antibody for acetylated tubulin (see Methods). We showed that regenerating legs could be fixed and immunofluorescently stained after long-term live imaging. The stained cells could then be identified in the live recordings and tracked (*Figure 6A, A'*).

In a non-conventional experimental animal like *Parhyale*, this approach is limited by the scarcity of antibodies that can serve as cell-type markers. Using single-nucleus transcriptional profiling, we previously identified approximately 15 transcriptionally distinct cell types in adult *Parhyale* legs (*Almazán et al., 2022*), including epidermis, muscle, neurons, haemocytes, and a number of still unidentified cell types. Numerous marker genes can be identified in these data, but generating and validating antibodies for these markers is a severe bottleneck. To overcome this problem, we developed an alternative approach for detecting these marker genes by in situ staining, using short nucleotide probes.

Numerous efforts to apply in situ hybridisation protocols on *Parhyale* legs have failed in the past, because the chitinous exoskeleton surrounding the adult legs is impermeable to probes or gives high unspecific staining. By fragmenting the legs and using long incubation times, the permeability problem was partly solved for antibodies (*Almazán et al., 2022*) but could not be overcome with conventional in situ hybridisation probes, presumably due to their large molecular size. We therefore turned to hybridisation chain reaction (HCR), a sensitive in situ hybridisation method that makes use of much shorter probes (*Choi et al., 2018*). Starting from an HCR protocol used in *Parhyale* embryos (*Bruce et al., 2021*), we adapted HCR to adult legs by extending the permeabilisation step, adding a decalcification step to soften the cuticle, adjusting probe concentrations and extending the incubation and washing steps (see Methods). Using this method, we are able to label specific populations of cells, such as neurons (using probes for *futsch*, *Hummel et al., 2000*) and mechanosensory organs (using probes for *nompA*, *Chung et al., 2001*), in late embryos and adult legs (*Figure 6—figure supplement 1*). This is the first time we are able to apply an in situ hybridisation approach in adult legs, overcoming the cuticular permeability barrier.

As described earlier for immunostainings, we applied HCR to regenerated legs immediately after live imaging and were able to locate the stained cells in the live image recordings. This approach allows us to track the origins of molecularly identified cells during the course of regeneration. As a proof of principle, we tracked the origin of a population of *spineless*-expressing cells in the distal part of the regenerated carpus (*Figure 7*). In *Drosophila*, *spineless* is expressed in a ring of cells in the distal part of developing legs; it encodes a transcription factor that plays a role in patterning of the tarsal segments (*Duncan et al., 1998*; *Natori et al., 2012*). We find that the *spineless* orthologue of *Parhyale* is expressed in a similar pattern at the distal end of the regenerated carpus (*Figure 7—figure supplement 1*). Cell tracking reveals that the *spineless*-expressing cells of the distal carpus originate from epidermal cells that were widely dispersed in the carpus at the time of amputation (*Figure 7*). These epidermal progenitors undergo 0, 1, or 2 cell divisions and generate mostly *spineless*-expressing cells (*Figure 7—figure supplement 1*).

## Future directions

*Parhyale* stands out as an experimental system in which we can image the entire process of leg regeneration at single-cell resolution and follow the lineages and the fates of cells as the leg is re-made. This provides unique opportunities to identify the progenitors of regenerated tissues, their degree of commitment, locations, division patterns and behaviours during regeneration. Combined with

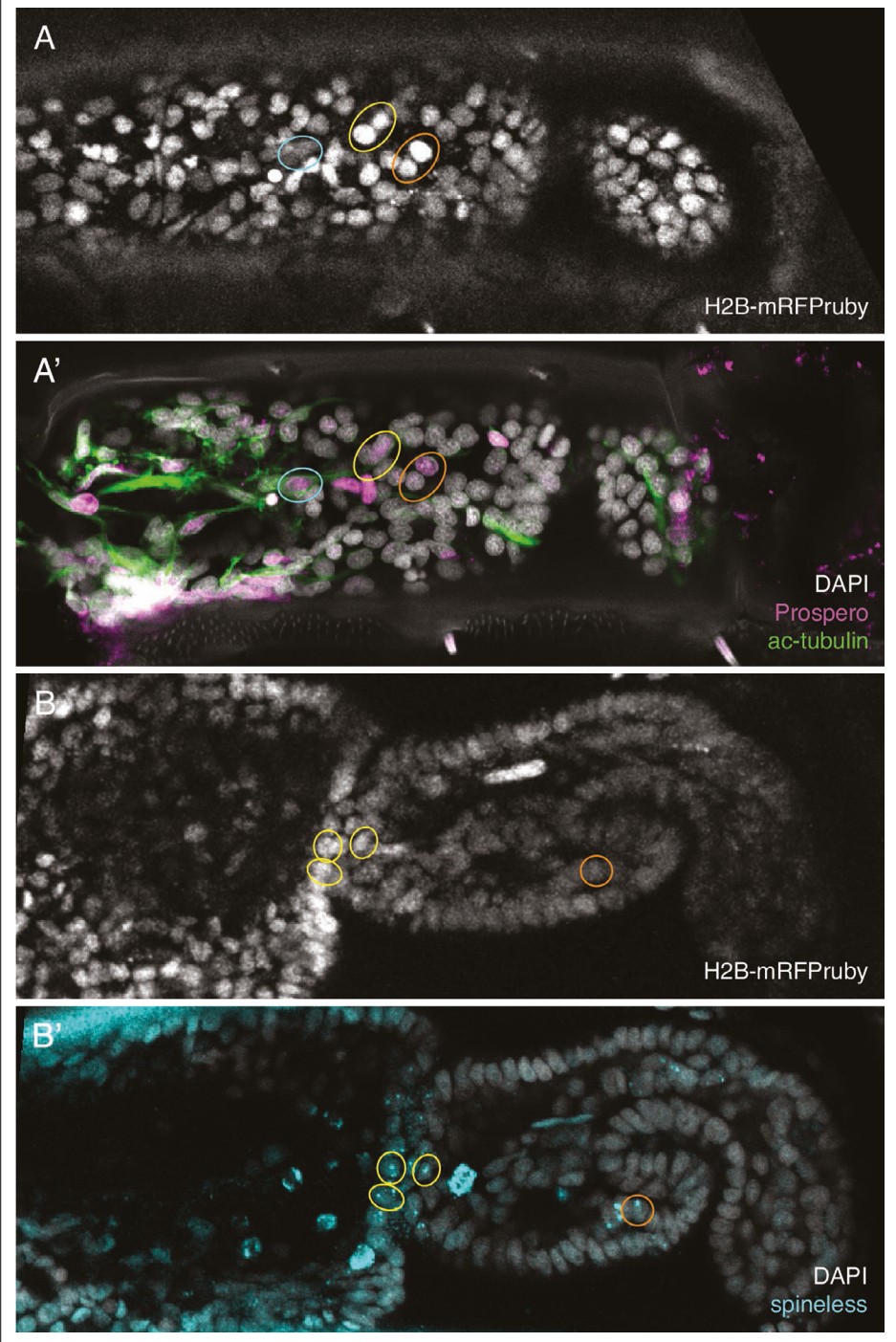

**Figure 6.** Identification of in situ stained cells in live image recordings. (A,A') Regenerated T5 leg at 189 hpa, in the last frame of a live recording (**A**) and after immunostaining with antibodies for Prospero and acetylated alpha tubulin (**A'**). (B,B') Regenerated T5 leg imaged at 97 hpa, in the last frame of a live recording (**B**) and after hybridisation chain reaction (HCR) with probes for orthologue of *spineless* mostly nuclear dots corresponding to nascent transcripts, (**B'**). The same cells can be identified in the live recordings and in immunofluorescence or HCR stainings, as highlighted by coloured circles. Note that not all corresponding cells are visible in these optical sections, due to slight differences in mounting and tissue distortion.

The online version of this article includes the following figure supplement(s) for figure 6:

**Figure supplement 1.** In situ staining of *Parhyale* legs by hybridisation chain reaction (HCR).

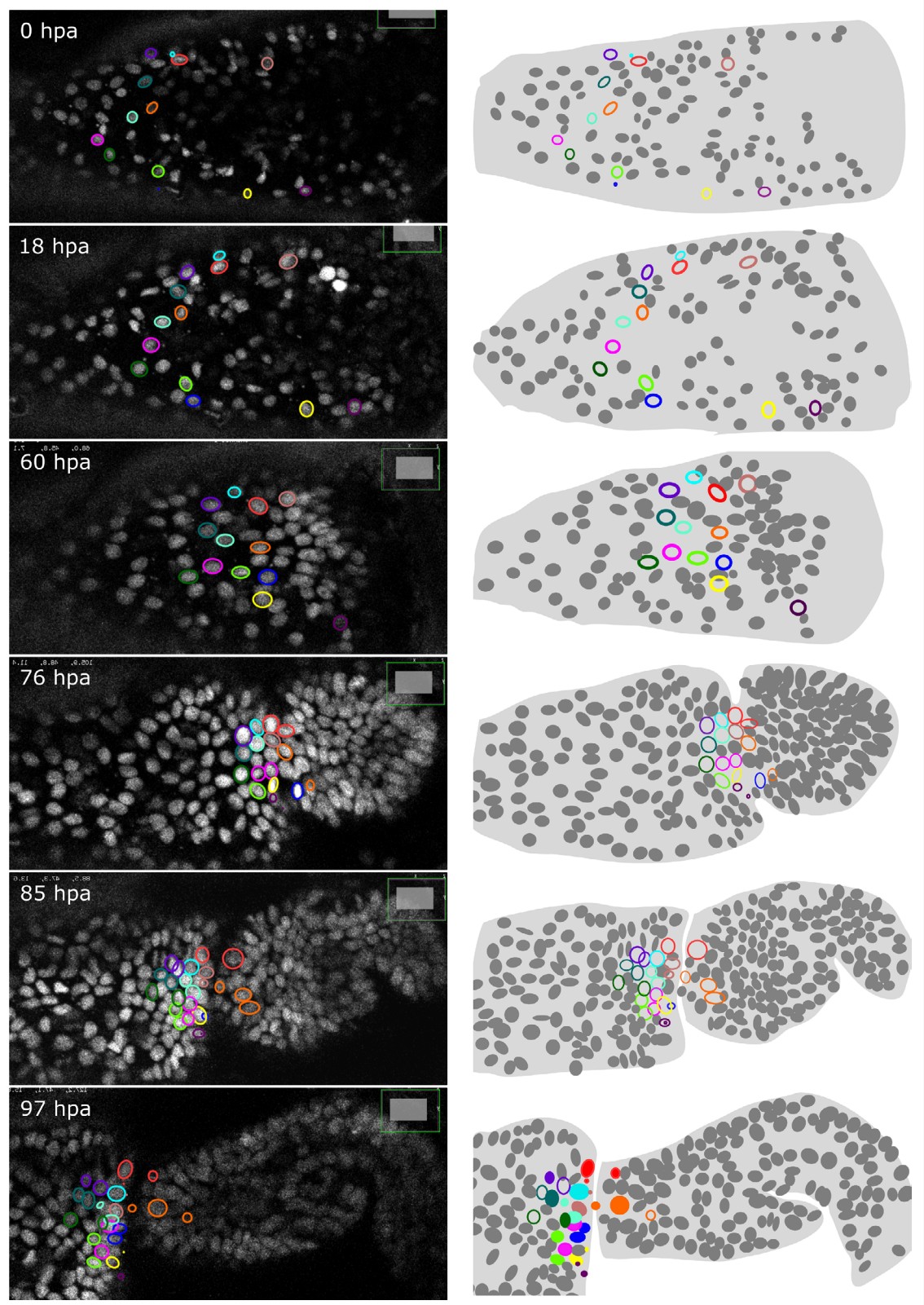

**Figure 7.** Tracking the progenitors of *spineless*-expressing cells in the distal carpus. Snapshots of live recording of a regenerating *Parhyale* T5 leg at 0, 18, 60, 76, 85, and 97 h post amputation (left) and corresponding illustrations of the same legs (right). Coloured circles highlight cells that contribute to *spineless*-expressing cells in the distal part of the carpus; each colour highlights the lineage of a distinct progenitor cell. In the bottom-right panel, *spineless*-expressing nuclei (identified by HCR) are marked by filled circles, whereas *spineless*-non-expressing nuclei derived from the same progenitors

*Figure 7 continued on next page*

*Figure 7 continued*
are marked by open circles. The images show single optical sections as they appear in the Mastodon user interface; nuclei that are only partly captured in the current optical section appear as smaller circles. Distal parts of the leg oriented towards the right.

The online version of this article includes the following figure supplement(s) for figure 7:

**Figure supplement 1.** *Spineless* expression in a regenerating *Parhyale* leg.

high-resolution live imaging of developing embryonic legs (*Wolff et al., 2018*), it also offers the chance to compare the cellular dynamics of leg regeneration with those of leg development.

Currently, there are two major limitations in this research approach. First, image quality limits the efficiency with which we can track cells, particularly in the deeper tissue layers. Improved image quality could be achieved with higher numerical aperture objectives, stronger fluorescence signals and longer exposure times, to increase signal-to-noise ratios and resolution. These improvements, however, come with trade-offs in terms of increased light exposure and tissue damage. An obvious improvement, in this respect, would come from replacing confocal microscopy by a more gentle imaging method, such as light-sheet microscopy (*Stelzer et al., 2021*). Unfortunately, our current method for immobilising *Parhyale* legs for live imaging is incompatible with conventional light-sheet microscopy, because it requires glueing and imaging the specimens through a glass coverslip. At present, the method we describe here provides the most robust approach to image leg regeneration in *Parhyale*.

A second limitation is the dearth of fluorescent markers that we can use to identify different cell types and cellular transitions during the course of live imaging. Until we develop such markers, the in situ labelling approaches described here will provide an efficient way to identify different cell fates at the end of live imaging.

## Materials and methods

### *Parhyale* culture and transgenic lines

Wild-type and transgenic lines of *P. hawaiensis* were bred as described before (*Browne et al., 2005*; *Paris et al., 2022*). For live imaging, we used transgenic animals carrying stable genomic integrations of the *pMinos[3xP3-DsRed; PhHS-H2B-mRFPruby]* transgene (*Alwes et al., 2016*; *Wolff et al., 2018*). The animals were in some cases kept in the dark for 1–4 weeks prior to live imaging, to prevent the build-up of algae on the cuticle.

### Live imaging

To induce H2B-mRFPruby expression, animals were heat-shocked for 45 min at 37°C, 12–24 h prior to the amputation. Prior to amputation, the animals were anaesthetised using 0.02% clove oil (in artificial seawater) and mounted on the glass surface of 35 mm Ibidi Glass Bottom Dishes (Ibidi μ-Dish 35 mm, #81158), which served as the microscope cover slip, using surgical glue (2-octyl-cyanoacrylate, Derm-abond Mini #AHVM12), as described previously (*Alwes et al., 2016*). Aliquots of the glue are stored in the dark at room temperature. T4 and/or T5 limbs were amputated at the distal end of the carpus using a microsurgical knife (Fine Science Tools #10316-14).

Live imaging was performed on a Zeiss LSM 800 confocal microscope equipped with a Plan-Apochromat 20×/0.8 M27 objective (Zeiss 420650-9901-000) and a temperature control chamber set to 26 or 27°C (with the exception of dataset li51, imaged at 29°C). Imaging was carried out with the 561 nm excitation laser, a pinhole set to 38 μm (1 Airy unit), an emission window of 400–700 nm, and a GaAsP detector (detector gain 700–750 V, detector offset 0, digital gain 1.0). Laser power was typically set to 0.3–0.8%, which yields 0.51–1.37 μW at 561 nm (measured with a ThorLabs Microscope Slide Power Sensor, #S170C).

Images were acquired with a pixel size of 0.31–0.62 μm and a speed of 0.76–2.06 μs per pixel, averaging 2–4 images per slice. These settings were determined by early tests in which we scanned the same specimen with a range of scan speed and averaging settings and measured the signal-to-noise and contrast ratios as described in *Ulman et al., 2017* (see *Figure 2—figure supplement 1*).

Image stacks typically consisted of 8–36 *z* steps taken with a step size of 1.24–2.48 μm. Stacks were acquired at 10- to 20-min time intervals, over up to 330–1030 time points, using the Definite Focus feature of the microscope to compensate for axial image drift. Image size was typically

1024x800 pixels and image depth was 8-bit. The table in Supplementary Data 1, available at https://doi.org/10.5281/zenodo.15181497, provides more information on individual imaging experiments. Under these conditions, approximately 85% of animals (18 out of 21) survive the mounting and imaging procedure. Among the imaged legs, approximately 20% show tissue damage/retraction (5 out of 37) or become arrested before the onset of regeneration (2 out of 37), while 80% go on to regenerate normally.

To image multiple regenerating legs simultaneously (e.g. li24, li41, li48, li51, and li52 in *Figure 3*), T4 and T5 limbs on the same side of the animal were positioned parallel to each other during mounting and imaged using the 'tile' imaging option of the Zeiss Blue software; the two limbs were imaged as individual tiles.

Heat shocks were carried out during the course of imaging to boost H2B-mRFPruby expression by setting the temperature of the stage (Pecon Heatable Universal Mounting Frame KH-R S1, Pecon #020-800 153) to 40°C for 2 hr; the temperature setting of the microscope incubation chamber was kept at 26 or 27°C. At the end of the heat shock, the temperature control of the stage was turned off. Typically, 1–3 heat shocks are required during the time course of imaging.

After image acquisition, stacks were exported in .czi format and post-processed in Fiji (*Schindelin et al., 2012*). Image stacks were concatenated in Fiji (Stacks/Tools/Concatenate function), *xy* image shifts caused by heat shock were corrected using the *align-slices3d* Fiji plug-in (https://github.com/elephant-track/align-slices3d; *Sugawara, 2025a*; https://doi.org/10.5281/zenodo.15162844) and *z* shifts were corrected manually in Fiji, as described previously (*Sugawara et al., 2022*).

Detecting cell divisions: Cell divisions in *Figure 4* were detected in two ways. The divisions in li13-t4 were extracted from the extensive cell tracking carried out previously in that dataset using Elephant and manual proofreading (*Sugawara et al., 2022*). The divisions in the other datasets were detected by training an Elephant detection model to identify the nuclei of dividing cells at metaphase, followed by manual proofreading of the predictions. Training was conducted on cropped volumes of size (96 × 96 × 96 pixels) extracted from the resized image volumes that were rescaled by a factor of eight along the *Z*-axis to render them nearly isotropic. Each volume was augmented with random flips along all three dimensions to enhance the robustness of the model. In each iteration of training, the model was trained over 10 epochs with a batch size of 20, using the Adam optimizer with a learning rate of 0.0001. Prediction was performed on the resized image volumes using patches of size (96 × 96 × 96 pixels). A threshold of 0.8 was applied to the probabilities to filter out low-confidence predictions. The parameters for the centre detection were set to: minimum radius (rmin) 1 μm, maximum radius (rmax) 5 μm, and suppression distance (dsup) 3 μm (see *Sugawara et al., 2022*). This approach captures the temporal pattern of cell divisions in the regenerating legs, but is not exhaustive. The results of this analysis are provided in Supplementary Data 9 available at https://doi.org/10.5281/zenodo.15181497.

## Cell tracking

Cells were tracked on Mastodon (https://github.com/mastodon-sc/mastodon), implemented as a plug-in in Fiji (*Schindelin et al., 2012*), either manually or using the Elephant plug-in (*Sugawara et al., 2022*). A 'backtracking' function was added to Elephant (v.0.4.3, released 9 June 2023), allowing users to track cells of particular interest. This function allows users to automatically trace the position of a selected cell backward in time using the flow estimation algorithm integrated in Elephant. Prior to initiating the tracking process, a detection model and a flow estimation model are trained on the target image data (or a similar dataset) to ensure precise tracking. Users can select the target cell in the Elephant graphical user interface. The method then processes each time point iteratively, creating track links by predicting the target cell's position at the previous time point and identifying the closest detected cell. This is achieved by predicting spots around the cell's estimated position. If no cells are detected in the vicinity, the algorithm interpolates a new spot at the position estimated by the flow model. The maximum number of time points for interpolation can be specified. For detailed information on the training of the detection and flow estimation models, please refer to *Sugawara et al., 2022*. The implementation of the backtracking function can be found in https://github.com/elephant-track/elephant-client/blob/main/src/main/java/org/elephant/actions/BackTrackAction.java (*Sugawara, 2025b*; https://doi.org/10.5281/zenodo.15162848).

## Optimising trade-offs between imaging resolution, image quality and light exposure

To optimise our image acquisition strategy, we acquired a 3D image dataset using the setup described in the previous section, at higher resolutions and image quality than in our standard experiments (0.31 × 0.31 µm pixel size in *xy*, acquired at a speed of 1.03 µs per pixel, 1.24 µm step size in *z*, 10-min time intervals, with 4 averaged image replicates per slice). We then sub-sampled this dataset in different ways, yielding five reduced datasets that differ in *xy* resolution, *z* resolution and image quality (datasets #1–5, see Table 1A). Dataset #1 was made by discarding every second *z* slice and every second time point, yielding resolutions and image quality similar to those of our typical imaging datasets. Dataset #2 was made by discarding every second time point and by averaging only 2 images per slice, instead of 4 (better *z* resolution, but worse image quality than dataset #1). Dataset #3 was made by discarding every second *z* slice and by averaging only 2 images per slice, instead of 4 (better temporal resolution, but worse image quality than dataset #1). Dataset #4 was made by discarding every second pixel in the *x* and *y* dimensions (better *z* and temporal resolutions, but lower *xy* resolutions than dataset #1). Dataset #5 was made by using a single image per *z* slice and time point, rather than by averaging four image replicates (better *z* and temporal resolutions, but lower image quality than dataset #1). Averaging operations were made by averaging the raw pixel values of two or four imaging replicates. These datasets were generated using a script available at https://github.com/ksugar/czi_sampling/blob/main/src/czi_sampling/sampling.py (*Sugawara, 2025c*; https://doi.org/10.5281/zenodo.15162855).

Using Mastodon, we generated ground truth annotations on the original image dataset, consisting of 278 cell tracks, including 13,888 spots and 13,610 links across 55 time points (see Supplementary Data 4 at https://doi.org/10.5281/zenodo.15181497). From these annotations, we selected 10 tracks (including 5 tracks with cell divisions and 5 tracks without) for training Elephant; the selected tracks consisted of 699 spots and 689 links.

The ground truth annotations were then applied to datasets #1–5. To apply these annotations to datasets #1 and #2, we had to reduce the number of time points. We did this using a Mastodon plug-in (available at https://github.com/mastodon-sc/mastodon-averoflab; *Sugawara, 2025d*; https://doi.org/10.5281/zenodo.15162873), generating a reduced set of ground truth annotations with 7068 spots and 6790 links. These annotations include 356 spots and 346 links in the 10 tracks selected for training Elephant.

To ensure a fair comparison of the tracking metrics across datasets of varying sizes, the training of the models was conducted using resized volumes, standardised to the same dimensions as the highest resolution data (512 × 512 × 20 pixels). Additionally, the number of iteration steps during training was kept nearly identical across all datasets, ensuring consistent model training and allowing for a reliable comparison of tracking performance.

The detection models were trained using a standardised approach to ensure consistency across the datasets. For datasets #1, #3, and #4, the input image volumes were resized to dimensions of (512 × 512 × 20 pixels) with nearest-neighbour interpolation. For the remaining datasets, which were already sized at (512 × 512 × 20 pixels), no resizing was necessary. Training was conducted on cropped volumes of size (128 × 128 × 16 pixels) extracted from the resized image volumes. Each volume was augmented with random flips along all three dimensions to enhance the robustness of the model. The models were trained over 10 epochs with a batch size of 20, using the Adam optimizer with a learning rate of 0.01. For datasets #1 and #2, which contain 28 time points with annotations, 100 volumes were cropped from each time point for training in each epoch. For the other datasets, which contain 55 time points with annotations, 50 volumes were cropped from each time point for training in each epoch. During training, the centre ratio was set to 0.4, and the background threshold was set to 0, meaning that only explicitly annotated voxels were utilised in the training process. The source code used for training the detection models is available at https://github.com/elephant-track/elephant-server/blob/trackathon-paper/script/train_trackathon.sh (*Sugawara, 2025e*; https://doi.org/10.5281/zenodo.15161691). The trained detection models were then used to predict nuclei in datasets #1–5, following the same procedure across all datasets. The input image volumes were resized in the same manner as during training, ensuring that the model operated on images of dimensions (512 × 512 × 20 pixels). Prediction was performed directly on the resized image volumes without further patching, allowing the model to process the entire volume in one step. A threshold of 0.7 was applied to the

probabilities to filter out low-confidence predictions. The parameters for the centre detection were set to: minimum radius ($r_{min}$) 1 μm, maximum radius ($r_{max}$) 5 μm, and suppression distance ($d_{sup}$) 3 μm (see *Sugawara et al., 2022*).

The flow models were trained following a similar preprocessing as described for the detection models. Input image volumes were resized and cropped with random flips across all dimensions, as in the training of detection models. The flow models were trained over 10 epochs with a batch size of 1, employing the Adam optimizer with a learning rate of 0.00001. For datasets #1 and #2, which include 27 pairs of successive time points with annotations, 100 volumes were cropped from each time point for training in each epoch. For the other datasets, which include 54 pairs of successive time points with annotations, 50 volumes were cropped from each time point for training in each epoch. The source code used for training the flow models is available at https://github.com/elephant-track/elephant-server/blob/trackathon-paper/script/train_trackathon_flow.sh (*Sugawara, 2025e*). The trained flow models were then employed for tracking by providing flow predictions. To ensure a fair comparison of the tracking metrics across datasets of varying sizes, the input image volumes were resized in the same manner as during training, ensuring that the model operated on images with dimensions of (512 × 512 × 20 pixels). Tracking was performed directly on the resized image volumes without additional patching, allowing the model to process the entire volume in a single step. In the nearest neighbour linking algorithm, the maximum distance threshold of 5 μm was applied (see *Sugawara et al., 2022*). Additionally, the maximum number of links per cell was set to 2, enabling the tracking of dividing cells.

The tracking results were evaluated using several metrics to comprehensively assess performance under each condition. Detection and tracking accuracy (DET and TRA) were defined in the Cell Tracking Challenge (*Matula et al., 2015*; *Ulman et al., 2017*); DET was used to evaluate the model's ability to correctly identify the locations of cells, and TRA to assess the overall performance of the tracking model, including the accuracy of both detection and linking.

Detection precision and recall were calculated based on the DET analysis, offering insights into the model's ability to correctly identify true positives (*precision = TP/(TP + FP)*) and its capability to recall all relevant instances (*recall = TP/(TP + FN)*). Linking precision and recall was derived in the same manner from the TRA analysis. In order to measure specifically the linking performance, independently of detection, we measured the linking precision and recall from tracking performed on the spots of the ground truth data (TRA on GT, *Table 1*). DET and TRA scores were normalised to the range [0, 1] by calculating the cost of transforming the computed set of detections and links into the reference set (ground truth data). This normalisation provides a representation of model performance that is comparable across different datasets, independent of their size.

Last, we estimated the time or effort that would be required to proofread the tracking results (in arbitrary units) based on the acyclic oriented graphs matching metric, which is part of the calculation of TRA scores (see Table 6 in *Matula et al., 2015*). This metric depends both on the model's performance and the total number of spots and links across all time points; as the number of time points increases, the number of data points requiring correction also increases, thereby impacting the proofreading time.

To account for variation in the training, three detection and three linking (flow) models were trained and evaluated independently on each dataset. *Table 1* shows the averaged scores of the three iterations. The source code used for the evaluation is available at https://github.com/ksugar/long-term-live-imaging (*Sugawara, 2025f*) and https://doi.org/10.5281/zenodo.15162928.

Image deconvolution and denoising: To test the impact of image deconvolution on cell tracking performance, we applied the Huygens Essential 21.04 software (Scientific Volume Imaging), using the Standard Express Deconvolution algorithm, to dataset #1. To test the impact of denoising, we applied the Noise2Void denoising tool (*Krull et al., 2018*) with the settings given in https://github.com/ksugar/n2v/blob/czi/examples/3D/03_training_20220927.ipynb and https://github.com/ksugar/n2v/blob/czi/examples/3D/04_prediction_20220927.ipynb to dataset #1 (*Sugawara, 2025g*; https://doi.org/10.5281/zenodo.15162863). We then tested the cell tracking performance of Elephant on the deconvoluted and denoised datasets, as described for dataset #1 in the previous section.

## Immunostaining

After live imaging, animals were fixed while attached to the glass surface that was used for imaging, using 4% paraformaldehyde (20% EM grade stock solution, Electron Microscopy Sciences #15713)

in filtered artificial seawater for 20 min at room temperature. The animals were washed three times, 15 min each, in 1× PBS with 0.1% Triton X-100. Using insect pins, holes were made in the melanised scab of the leg stumps. The regenerated legs were then detached from the glass surface by cutting them out at the distal part of the merus. They were then incubated for 1 h at room temperature in 1× PBS with 0.1% Triton X-100, 0.1% sodium deoxycholate, 0.1% bovine serum albumin, and 1% normal goat serum, and stained as described previously (*Almazán et al., 2022*). As primary antibodies, we used an antibody raised against the *Parhyale* Repo protein (*Almazán et al., 2022*), an antibody raised against the Prospero protein of a butterfly, which likely cross-reacts with the Prospero protein in *Parhyale* (*Perry et al., 2016*), and monoclonal antibody 6-11B-1, which recognises acetylated alpha tubulin (diluted 1:1000; Sigma-Aldrich T6793, RRID:AB_477585).

## Hybridisation chain reaction

The HCR protocol for adult *Parhyale* legs is based on previous protocols developed for chicken embryos (*Choi et al., 2018*), *Parhyale* embryos (*Bruce et al., 2021*) and crinoids (*Gattoni, 2022*). Adult legs were collected and fixed in 4% paraformaldehyde (20% EM grade stock solution, Electron Microscopy Sciences #15713) in filtered artificial seawater for 50 min at room temperature. The legs were then washed at least three times (10 min each) in PTw (1× PBS, 0.1% Tween-20). At this stage, the legs can be stored by washing in increasing concentrations of methanol (50%, 70%, and 90% methanol in PTw) before transferring to 100% methanol at –20°C. The samples can be stored in this way for several months and recovered by washing in decreasing concentrations of methanol (75%, 50%, and 25% in PTw), and then 3 times in PTw (10 min each). The transfer to methanol and storage are optional.

The legs were then cut into fragments comprising one or more podomeres and placed in 500 µl of decalcification solution (5% EDTA in DEPC-treated water, pH adjusted to 7.5 using NaOH) overnight at room temperature.

On the following day, the leg fragments were transferred to 500 µl of detergent solution (1% SDS, 0.5% Tween-20, 50 mM Tris-HCl pH 7.5, 150 mM NaCl, in DEPC-treated water) for 2 hr. They were then washed briefly in PTw (10 min), re-fixed in 4% paraformaldehyde in PTw for 10 min. The fixative was removed with three 10 min washes in PTw. The leg fragments were transferred to 50 µl of Probe Hybridisation Buffer (30% formamide, 5 x SSC, 9 mM citric acid (pH 6.0), 0.1% Tween-20, 50 µg/ml heparin, 1 x Denhardt's solution, 10% dextran sulphate; supplied by Molecular Instruments) that had been pre-heated to 37 °C, and kept for 30 min at 37 °C. The probe solution was prepared by adding 0.8 pmol (0.8 µl of 1 µM stock solution) of each probe to 50 µl of pre-warmed Probe Hybridisation Buffer at 37°C (probe concentration can be increased up to two to three times to improve signal). The 50 µl probe solution was added to the 50 µl of Probe Hybridisation Buffer containing the leg fragments (to a final volume of 100 µl), mixed and incubated overnight at 37°C.

On the third day, the probe solution was recovered (can be kept at –20°C and re-used) and the leg fragments were processed through four 5 min and two 30 min washes at 37°C, in Probe Wash Buffer (30% formamide, 5× SSC, 9 mM citric acid (pH 6.0), 0.1% Tween-20, 50 µg/ml heparin; supplied by Molecular Instruments) that had been pre-heated to 37°C. The leg fragments were then washed twice (5 min each) with 500 µl of SSCT (5× SSC and 0.1% Tween-20 in DEPC-treated water) at room temperature. SSCT was removed, and the sample was incubated in 50 µl of Amplification Buffer pre-warmed to room temperature (5× SSC, 0.1% Tween-20, 10% dextran sulphate; supplied by Molecular Instruments), for 30 min at room temperature. During this step, 2 µl of each hairpin (3 µM stock solutions of hairpins h1 and h2, supplied by Molecular Instruments, see below) were placed in separate 0.5 µl Eppendorf tubes, incubated at 95°C for 90 s, and then allowed to cool down to room temperature in the dark for 30 min. These hairpins were then mixed in 50 µl of Amplification Buffer, and mixed with the 50 µl of Amplification Buffer containing the leg fragments (to a final volume of 100 µl). This solution was incubated overnight at room temperature.

On the fourth day, the amplification solution containing the hairpins was recovered (can be kept at –20°C and re-used) and the leg fragments were processed through three 5 min washes and two 30 min washes in SSCT, at room temperature. The leg fragments were then transferred to 50% glycerol (in 1× PBS, pH 7.4) containing 1 µg/ml DAPI for 2 h or more, at room temperature, then to 70% glycerol (in 1× PBS), and mounted for imaging.

In this study, we used HCR probes for the *Parhyale* orthologues of *futsch* (MSTRG.441), *nompA* (MSTRG.6903) and *spineless* (MSTRG.197), ordered from Molecular Instruments (20 oligonucleotides

per probe set). The transcript sequences targeted by each probe set are given in Supplementary Data 6 at https://doi.org/10.5281/zenodo.15181497. These were detected using the amplifier sets B1-647 (for *futsch*) and B5-546 (for *nompA* and *spineless*). The stained legs were imaged on a Zeiss LSM 800 confocal microscope.

To perform HCR after live imaging, the imaged legs were detached from the glass surface by cutting the leg at the distal part of the merus, using a microsurgical knife (Fine Science Tools #10316-14). The leg fragments were then placed directly into 500 μl of 4% paraformaldehyde (20% EM grade stock solution, Electron Microscopy Sciences #15713) in filtered artificial seawater for 50 min at room temperature. We then followed the HCR protocol described above.

### Identifying in situ stained cells in live image data

After staining, the leg fragments were transferred to glycerol with DAPI (as described above) and mounted between two coverslips, to allow imaging from two sides. The natural shape of the carpus and the thin layer of glue still attached to the cuticle helped to flat mount these leg fragments and to image them from the same orientation as in live imaging. After imaging, the 3D image stacks of the in situ stained leg and of the last time point of live imaging were opened in Fiji and compared, to visually match the DAPI-stained nuclei with the H2B-mRFPruby-marked nuclei, respectively.

## Acknowledgements

We thank Stefan Hahmann for help in testing the Mastodon cell division grapher, Mathilde Paris for checking gene annotations for probe design, and past and present members of our team for valuable feedback. This research was supported by the European Research Council under the European Union Horizon 2020 programme (grant ERC-2015-AdG 694918 'ReLive' to MA), by the Agence Nationale de la Recherche of France (grant ANR-21-CE13-0044-01 'DeepLineage' to MA), by a doctoral fellowship from Boehringer Ingelheim Fonds to ÇÇ, and by the Core Research for Evolutionary Science and Technology (CREST) programme of Japan (grant JPMJCR1926 to KS).

## Additional information

### Competing interests

Ko Sugawara: KS is employed part-time by LPIXEL Inc. The other authors declare that no competing interests exist.

### Funding

| Funder | Grant reference number | Author |
|---|---|---|
| European Research Council | ERC-2015-AdG 694918 | Michalis Averof |
| Agence Nationale de la Recherche | ANR-21-CE13-0044-01 | Michalis Averof |
| Boehringer Ingelheim Fonds | | Çağrı Çevrim |
| Core Research for Evolutionary Science and Technology | JPMJCR1926 | Ko Sugawara |

The funders had no role in study design, data collection, and interpretation, or the decision to submit the work for publication.

### Author contributions

Çağrı Çevrim, designed the experiments, performed the live imaging, performed the cell tracking, developed methods combining live imaging with immunostaining, prepared the data and figures, edited the manuscript; Béryl Laplace-Builhé, designed the experiments, performed the live imaging, performed the cell tracking, developed methods combining live imaging with HCR, prepared the

data and figures, edited the manuscript; Ko Sugawara, designed the experiments, performed the cell tracking, developed Elephant, performed denoising and data subsampling, measured tracking performance, prepared the data and figures, edited the manuscript; Maria Lorenza Rusciano, developed the HCR method; Nicolas Labert, performed the cell tracking; Jacques Brocard, provided imaging expertise and performed the image deconvolution; Alba Almazán, developed the HCR method; Michalis Averof, conceived and supervised the project, designed the experiments, wrote the manuscript, prepared the data and figures, edited the manuscript

## Author ORCIDs
Çağrı Çevrim https://orcid.org/0000-0002-4720-7944
Béryl Laplace-Builhé http://orcid.org/0000-0002-1437-587X
Ko Sugawara https://orcid.org/0000-0002-1392-9340
Maria Lorenza Rusciano https://orcid.org/0009-0000-6428-2674
Nicolas Labert https://orcid.org/0009-0007-2738-0532
Jacques Brocard https://orcid.org/0000-0002-0752-5737
Alba Almazán https://orcid.org/0000-0001-9933-4278
Michalis Averof https://orcid.org/0000-0002-6803-7251

Reviewer #1 (Public review): https://doi.org/10.7554/eLife.107534.2.sa1
Reviewer #2 (Public review): https://doi.org/10.7554/eLife.107534.2.sa2
Author response https://doi.org/10.7554/eLife.107534.2.sa3

## Additional files

### Supplementary files
MDAR checklist

### Data availability
All data files are available at the Zenodo public data repository, at this link: https://doi.org/10.5281/zenodo.15181497.

The following dataset was generated:

| Author(s) | Year | Dataset title | Dataset URL | Database and Identifier |
|---|---|---|---|---|
| Cevrim C, Laplace-Builhé B, Sugawara K, Averof M | 2025 | Long-term live imaging, cell identification and cell tracking in regenerating crustacean legs | https://doi.org/10.5281/zenodo.15181497 | Zenodo, 10.5281/zenodo.15181497 |

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
