## [Editor Report · eLife Assessment]

This study presents a **valuable** technical advance in the long-term live imaging of limb regeneration at cellular resolution in *Parhyale hawaiensis*. The authors develop and carefully validate a method to continuously image entire regenerating legs over several days while minimizing photodamage and optimizing conditions for robust cell tracking, together with post-hoc in situ identification of cell types. The data are **convincing**, the methodology is rigorous and clearly documented, and the results will be of interest to researchers in regeneration biology, developmental biology, and advanced live imaging and cell tracking software development.

[Editors' note: this paper was reviewed by Review Commons.]

---

## [Referee Report · Reviewer #1 (Public review)]

Building upon their previous work, the authors present an enhanced method for confocal live imaging of leg regeneration in the crustacean Parhyale hawaiensis. Parhyale is an emerging and tractable model system that offers insights into the evolution and mechanisms of development and regeneration. Çevrim et al. demonstrate the ability to image the complete leg regeneration process, spanning several days, with 10-20 minute time intervals and cellular resolution. They have concurrently optimized imaging conditions to enable cell tracking while minimizing phototoxicity. Additionally, they report successfully implementing HCR in situ hybridization in Parhyale, allowing for specific gene transcript staining at the endpoint of live imaging. This opens the possibility of assigning molecular identities to tracked cells.

A key challenge in many regeneration models is achieving continuous imaging throughout the entire regenerative process, as many organisms are difficult to immobilize or cannot tolerate extended imaging without stress. This manuscript's major strength lies in providing practical solutions to these challenges in Parhyale, a compelling and accessible arthropod model for limb regeneration. The authors also employ complementary tools to analyze time-lapse movies and correlate them with endpoint staining. Together, these advances will serve as a useful resource for researchers studying regeneration in Parhyale or in other systems where parts of this workflow can be adapted.

While the data demonstrating the methodological advancement and technical feasibility are solid, much of the benchmarking and regeneration characterization remains qualitative. This does not undermine the validity of the proof-of-principle, but limits the study's broader appeal.

---

## [Referee Report · Reviewer #2 (Public review)]

The manuscript by Çevrim et al. presents a live-imaging workflow that captures the complete leg regeneration process in the crustacean Parhyale hawaiensis, at a resolution suitable for cell tracking and gene expression analysis. Building on earlier work describing selective stages of leg regeneration (Alwes et al., 2016), the authors recorded 22 confocal time-lapse movies, starting from amputation to full regeneration. They defined three distinct phases of regeneration (wound closure, cell proliferation and morphogenesis, and differentiation) based on cellular and morphological features.

One movie was used to assess how imaging parameters (z-spacing, time intervals, and image quality) influence tracking reliability and the time required for manual proofreading, with an effort to minimize phototoxicity. Tracking was performed in the upper tissue layers using an improved version of the Mastodon plug-in Elephant in Fiji. The same sample was fixed post-imaging for in situ hybridization using an HCR protocol adapted for adult legs, targeting the gene spineless. This enabled the alignment of gene expression with specific cell lineages and the identification of progenitor cells present at the time of amputation.

In summary, the study provides a proof-of-principle for combining long-term live imaging, cell tracking, and gene expression analysis during regeneration. Given the labor-intensive nature of tracking over a 5-10 day time-lapse movie, the use of a single movie for this study is well justified. The workflow, from imaging to lineage reconstruction and molecular annotation, is successfully demonstrated and well documented with this dataset.

Although the biological insights from the cell lineage and molecular mapping are still limited, the methodology offers significant potential in regenerative biology to uncover the cellular and molecular contributions to tissue and cell type re-formation.

Confocal microscopy was used for live imaging, which restricted imaging to the upper 30 µm tissue layer. Light-sheet microscopy could have provided gentler imaging and enabled imaging from multiple angles to image the whole leg. While the authors acknowledge this possibility in the manuscript, they discarded it due to incompatibility between their mounting strategy and available light-sheet microscopes. As a future direction, optimizing the mounting approach for compatibility with light-sheet microscopes could enable more comprehensive tissue imaging.

---

## [Author Response]

We thank the reviewers for their feedback on our paper. We have taken all their comments into account in revising the manuscript. We provide a point-by-point response to their comments, below.

**Reviewer #1:**
Major comments:The manuscript is clearly written with a level of detail that allows others to reproduce the imaging and cell-tracking pipeline. Of the 22 movies recorded one was used for cell tracking. One movie seems sufficient for the second part of the manuscript, as this manuscript presents a proof-of-principle pipeline for an imaging experiment followed by cell tracking and molecular characterisation of the cells by HCR. In addition, cell tracking in a 5-10 day time-lapse movie is an enormous time commitment.My only major comment is regarding "Suppl_data_5_spineless_tracking". The image file does not load.It looks like the wrong file is linked to the mastodon dataset. The "Current BDV dataset path" is set to "Beryl_data_files/BLB mosaic cut movie-02.xml", but this file does not exist in the folder. Please link it to the correct file.

We have corrected the file path in the updated version of Suppl. Data 5.

Minor comments:The authors state that their imaging settings aim to reduce photo damage. Do they see cell death in the regenerating legs? Is the cell death induced by the light exposure or can they tell if the same cells die between the movies? That is, do they observe cell death in the same phases of regeneration and/or in the same regions of the regenerating legs?

Yes, we observe cell death during Parhyale leg regeneration. We have added the following sentence to explain this in the revised manuscript: "During the course of regeneration some cells undergo apoptosis (reported in Alwes et al., 2016). Using the H2B-mRFPruby marker, apoptotic cells appear as bright pyknotic nuclei that break up and become engulfed by circulating phagocytes (see bright specks in Figure 2F)."

We now also document apoptosis in regenerated legs that have not been subjected to live imaging in a new supplementary figure (Suppl. Figure 3), and we refer to these observations as follows: "While some cell death might be caused by photodamage, apoptosis can also be observed in similar numbers in regenerating legs that have not been subjected to live imaging (Suppl. Figure 3)."

Based on 22 movies, the authors divide the regeneration process into three phases and they describe that the timing of leg regeneration varies between individuals. Are the phases proportionally the same length between regenerating legs or do the authors find differences between fast/slow regenerating legs? If there is a difference in the proportions, why might this be?

Both early and late phases contribute to variation in the speed of regeneration, but there is no clear relationship between the relative duration of each phase and the speed of regeneration. We now present graphs supporting these points in a new supplementary figure (Suppl. Figure 2).

To clarify this point, we have added the following sentence in the manuscript: "We find that the overall speed of leg regeneration is determined largely by variation in the speed of the early (wound closure) phase of regeneration, and to a lesser extent by variation in later phases when leg morphogenesis takes place (Suppl. Figure 2 A,B). There is no clear relationship between the relative duration of each phase and the speed of regeneration (Suppl. Figure 2 A',B')."

Based on their initial cell tracing experiment, could the authors elaborate more on what kind of biological information can be extracted from the cell lineages, apart from determining which is the progenitor of a cell? What does it tell us about the cell population in the tissue? Is there indication of multi- or pluripotent stem cells? What does it say about the type of regeneration that is taking place in terms of epimorphosis and morphallaxis, the old concepts of regeneration?

In the first paragraph of Future Directions we describe briefly the kind of biological information that could be gained by applying our live imaging approach with appropriate cell-type markers (see below). We do not comment further, as we do not currently have this information at hand. Regarding the concepts of epimorphosis and morphallaxis, as we explain in Alwes et al. 2016, these terms describe two extreme conditions that do not capture what we observe during Parhyale leg regeneration. Our current work does not bring new insights on this topic.

Page 5. The authors mention the possibility of identifying the cell ID based on transcriptomic profiling data. Can they suggest how many and which cell types they expect to find in the last stage based on their transcriptomic data?

We have added this sentence: "Using single-nucleus transcriptional profiling, we have identified approximately 15 transcriptionally-distinct cell types in adult Parhyale legs (Almazán et al., 2022), including epidermis, muscle, neurons, hemocytes, and a number of still unidentified cell types."

Page 6. Correction: "..molecular and other makers.." should be "..molecular and other markers.."

Corrected

Page 8. The HCR in situ protocol probably has another important advantage over the conventional in situ protocol, which is not mentioned in this study. The hybridisation step in HCR is performed at a lower temperature (37°C) than in conventional in situ hybridisation (65°C, Rehm et al., 2009). In other organisms, a high hybridisation temperature affects the overall tissue morphology and cell location (tissue shrinkage). A lower hybridisation temperature has less impact on the tissue and makes manual cell alignment between the live imaging movie and the fixed HCR in situ stained specimen easier and more reliable. If this is also the case in Parhyale, the authors must mention it.

This may be correct, but all our specimens were treated at 37°C, so we cannot assess whether hybridisation temperature affects morphological preservation in our specimens.

Page 9. The authors should include more information on the spineless study. What been is spineless? What do the cell lineages tell about the spineless progenitors, apart from them being spread in the tissue at the time of amputation? Do spineless progenitors proliferate during regeneration? Do any spineless expressing cells share a common progenitor cell?

We now point out that spineless encodes a transcription factor. We provide a summary of the lineages generating spineless-expressing cells in Suppl. Figure 6, and we explain that "These epidermal progenitors undergo 0, 1 or 2 cell divisions, and generate mostly spineless-expressing cells (Suppl. Figure 5)."

Page 10. Regarding the imaging temperature, the Materials and Methods state "... a temperature control chamber set to 26 or 27°C..."; however, in Suppl. Data 1, 26°C and 29°C are indicated as imaging temperatures. Which is correct?

We corrected the Methods by adding "with the exception of dataset li51, imaged at 29°C"

Page 10. Regarding the imaging step size, the Materials and Methods state "...step size of 1-2.46 µm..."; however, Suppl. Data 1 indicate a step size between 1.24 - 2.48 µm. Which is correct?

We corrected the Methods.

Page 11. Correct "...as the highest resolution data..." to "...at the highest resolution data..."

The original text is correct ("standardised to the same dimensions as the highest resolution data").

Page 11. Indicate which supplementary data set is referred to: "Using Mastodon, we generated ground truth annotations on the original image dataset, consisting of 278 cell tracks, including 13,888 spots and 13,610 links across 55 time points (see Supplementary Data)."

Corrected

p. 15. Indicate which supplementary data set is referred to: "In this study we used HCR probes for the Parhyale orthologues of futsch (MSTRG.441), nompA (MSTRG.6903) and spineless (MSTRG.197), ordered from Molecular Instruments (20 oligonucleotides per probe set). The transcript sequences targeted by each probe set are given in the Supplementary Data."

Corrected

Figure 3. Suggestion to the overview schematics: The authors might consider adding "molting" as the end point of the red bar (representing differentiation).

The time of molting is not known in the majority of these datasets, because the specimens were fixed and stained prior to molting. We added the relevant information in the figure legend: "Datasets li-13 and li-16 were recorded until the molt; the other recordings were stopped before molting."

Figure 4B': Please indicate that the nuclei signal is DAPI.

Corrected

Supplementary figure 1A. Word is missing in the figure legend: ...the image also shows weak…

Corrected

Supplementary Figure 2: Please indicate the autofluorescence in the granular cells. Does it correspond to the yellow cells?

Corrected

Video legend for video 1 and 2. Please correct "H2B-mREFruby" to "H2B-mRFPruby".

Corrected

**Reviewer #2:**
Major comments:MC 1. Given that most of the technical advances necessary to achieve the work described in this manuscript have been published previously, it would be helpful for the authors to more clearly identify the primary novelty of this manuscript. The abstract and introduction to the manuscript focus heavily on the technical details of imaging and analysis optimization and some additional summary of the implications of these advances should be included here to aid the reader.

This paper describes a technical advance. While previous work (Alwes et al. 2016) established some key elements of our live imaging approach, we were not at that time able to record the entire time course of leg regeneration (the longest recordings were 3.5 days long). Here we present a method for imaging the entire course of leg regeneration (up to 10 days of imaging), optimised to reduce photodamage and to improve cell tracking. We also develop a method of in situ staining in cuticularised adult legs (an important technical breakthrough in this experimental system), which we combine with live imaging to determine the fate of tracked cells. We have revised the abstract and introduction of the paper to point out these novelties, in relation to our previous publications.

In the abstract we explain: "Building on previous work that allowed us to image different parts of the process of leg regeneration in the crustacean Parhyale hawaiensis, we present here a method for live imaging that captures the entire process of leg regeneration, spanning up to 10 days, at cellular resolution. Our method includes (1) mounting and long-term live imaging of regenerating legs under conditions that yield high spatial and temporal resolution but minimise photodamage, (2) fixing and in situ staining of the regenerated legs that were imaged, to identify cell fates, and (3) computer-assisted cell tracking to determine the cell lineages and progenitors of identified cells. The method is optimised to limit light exposure while maximising tracking efficiency."

The introduction includes the following text: "Our first systematic study using this approach presented continuous live imaging over periods of 2-3 days, capturing key events of leg regeneration such as wound closure, cell proliferation and morphogenesis of regenerating legs with single-cell resolution (Alwes et al., 2016). Here, we extend this work by developing a method for imaging the entire course of leg regeneration, optimised to reduce photodamage and to improve cell tracking. We also develop a method of in situ staining of gene expression in cuticularised adult legs, which we combine with live imaging to determine the fate of tracked cells."

MC 2. The description of the regeneration time course is nicely detailed but also very qualitative. A major advantage of continuous recording and automated cell tracking in the manner presented in this manuscript would be to enable deeper quantitative characterization of cellular and tissue dynamics during regeneration. Rather than providing movies and manually annotated timelines, some characterization of the dynamics of the regeneration process (the heterogeneity in this is very very interesting, but not analyzed at all) and correlating them against cellular behaviors would dramatically increase the impact of the work and leverage the advances presented here. For example, do migration rates differ between replicates? Division rates? Division synchrony? Migration orientation? This seems to be an incredibly rich dataset that would be fascinating to explore in greater detail, which seems to me to be the primary advance presented in this manuscript. I can appreciate that the authors may want to segregate some biological findings from the method, but I believe some nominal effort highlighting the quantitative nature of what this method enables would strengthen the impact of the paper and be useful for the reader. Selecting a small number of simple metrics (eg. Division frequency, average cell migration speed) and plotting them alongside the qualitative phases of the regeneration timeline that have already been generated would be a fairly modest investment of effort using tools that already exist in the Mastodon interface, I would roughly estimate on the order of an hour or two per dataset. I believe that this effort would be well worth it and better highlight a major strength of the approach.

The primary goal of this work was to establish a robust method for continuous long-term live imaging of regeneration, but we do appreciate that a more quantitative analysis would add value to the data we are presenting. We tried to address this request in three steps:

First, we examined whether clear temporal patterns in cell division, cell movements or other cellular features can be observed in an accurately tracked dataset (li13-t4, tracked in Sugawara et al. 2022). To test this we used the feature extraction functions now available on the Mastodon platform (see link). We could discern a meaningful temporal pattern for cell divisions (see below); the other features showed no interpretable pattern of variation.

Second, we asked whether we could use automated cell tracking to analyse the patterns of cell division in all our datasets. Using an Elephant deep learning model trained on the tracks of the li13-t4 dataset, we performed automated cell tracking in the same dataset, and compared the pattern of cell divisions from the automated cell track predictions with those coming from manually validated cell tracks. We observed that the automated tracks gave very imprecise results, with a high background of false positives obscuring the real temporal pattern (see images below, with validated data on the left, automated tracking on the right). These results show that the automated cell tracking is not accurate enough to provide a meaningful picture on the pattern of cell divisions.

Third, we tried to improve the accuracy of detection of dividing cells by additional training of Elephant models on each dataset (to lower the rate of false positives), followed by manual proofreading. Given how labour intensive this is, we could only apply this approach to 4 additional datasets. The results of this analysis are presented in Figure 4.

MC 3. The authors describe the challenges faced by their described approach:Using this mode of semi-automated and manual cell tracking, we find that most cells in the upper slices of our image stacks (top 30 microns) can be tracked with a high degree of confidence. A smaller proportion of cell lineages are trackable in the deeper layers.Given that the authors quantify this in Table 1, it would aid the reader to provide metrics in the manuscript text at this point. Furthermore, the metrics provided in Table 1 appear to be for overall performance, but the text describes that performance appears to be heavily depth dependent. Segregating the performance metrics further, for example providing DET, TRA, precision and recall for superficial layers only and for the overall dataset, would help support these arguments and better highlight performance a potential adopter of the method might expect.

In the revised manuscript we have added data on the tracking performance of Elephant in relation to imaging depth in Suppl. Figure 3. These data confirm our original statement (which was based on manual tracking) that nuclei are more challenging to track in deeper layers.

We point to these new results in two parts of the paper, as follows: "A smaller proportion of cells are trackable in the deeper layers (see Suppl. Figure 3)", and "Our results, summarised in Table 1A, show that the detection of nuclei can be enhanced by doubling the *z* resolution at the expense of *xy* resolution and image quality. This improvement is particularly evident in the deeper layers of the imaging stacks, which are usually the most challenging to track (Suppl. Figure 3)."

MC 4. Performance characterization in Table 1 appears to derive from a single dataset that is then subsampled and processed in different ways to assess the impact of these changes on cell tracking and detection performance. While this is a suitable strategy for this type of optimization it leaves open the question of performance consistency across datasets. I fully recognize that this type of quantification can be onerous and time consuming, but some attempt to assess performance variability across datasets would be valuable. Manual curation over a short time window over a random sampling of the acquired data would be sufficient to assess this.

We think that similar trade-offs will apply to all our datasets because tracking performance is constrained by the same features, which are intrinsic to our system; e.g. by the crowding of nuclei in relation to axial resolution, or the speed of mitosis in relation to the temporal resolution of imaging. We therefore do not see a clear rationale for repeating this analysis. On a practical level, our existing image datasets could not be subsampled to generate the various conditions tested in Table 1, so proving this point experimentally would require generating new recordings, and tracking these to generate ground truth data. This would require months of additional work.

A second, related question is whether Elephant would perform equally well in detecting and tracking nuclei across different datasets. This point has been addressed in the Sugawara et al. 2022 paper, where the performance of Elephant was tested on diverse fluorescence datasets.

**Reviewer #3:**
Major comments:• The authors should clearly specify what are the key technical improvements compared to their previous studies (Alwes et al. 2016, Elife; Konstantinides & Averof 2014, Science). There, the approaches for mounting, imaging, and cell tracking are already introduced, and the imaging is reported to run for up to 7 days in some cases.

In Konstantinides and Averof (2014) we did not present any live imaging at cellular resolution. In Alwes et al. (2016) we described key elements of our live imaging approach, but we were never able to record the entire time course of leg regeneration. The longest recordings in that work were 3.5 days long.

We have revised the abstract and introduction to clarify the novelty of this work, in relation to our previous publications. Please see our response to comment MC1 of reviewer 2.

• While the authors mention testing the effect of imaging parameters (such as scanning speed and line averaging) on the imaging/tracking outcome, very little or no information is provided on how this was done beyond the parameters that they finally arrived to.

Scan speed and averaging parameters were determined by measuring contrast and signal-to-noise ratios in images captured over a range of settings. We have now added these data in Supplementary Figure 1.

• The authors claim that, using the acquired live imaging data across entire regeneration time course, they are now able to confirm and extend their description of leg regeneration. However, many claims about the order and timing of various cellular events during regeneration are supported only by references to individual snapshots in figures or supplementary movies. Presenting a more quantitative description of cellular processes during regeneration from the acquired data would significantly enhance the manuscript and showcase the usefulness of the improved workflow.

The events we describe can be easily observed in the maximum projections, available in Suppl. Data 2. Regarding the quantitative analysis, please see our response to comment MC2 of reviewer 2.

• Table 1 summarizes the performance of cell tracking using simulated datasets of different quality. However only averages and/or maxima are given for the different metrics, which makes it difficult to evaluate the associated conclusions. In some cases, only 1 or 2 test runs were performed.

The metrics extracted from each of the three replicates, per dataset, are now included in Suppl. Data 4.

We consistently used 3 replicates to measure tracking performance with each of the datasets. The "replicates" column label in Table 1 referred to the number of scans that were averaged to generate the image, not to the replicates used for estimating the tracking performance. To avoid confusion, we changed that label to "averaging".

• OPTIONAL: An imaging approach that allows using the current mounting strategy but could help with some of the tradeoffs is using a spinning-disk confocal microscope instead of a laser scanning one. If the authors have such a system available, it could be interesting to compare it with their current scanning confocal setup.

Preliminary experiments that we carried out several years ago on a spinning disk confocal (with a 20x objective and the CSU-W1 spinning disk) were not very encouraging, and we therefore did not pursue this approach further. The main problem was bad image quality in deeper tissue layers.

Minor comments:• The presented imaging protocol was optimized for one laser wavelength only (561 nm) - this should be mentioned when discussing the technical limitations since animals tend to react differently to different wavelengths. Same settings might thus not be applicable for imaging a different fluorescent protein.

In the second paragraph of the Results section, we explain that we perform the imaging at long wavelengths in order to minimise photodamage. It should be clear to the readers that changing the excitation wavelength will have an impact for long-term live imaging.

• For transferability, it would be useful if the intensity of laser illumination was measured and given in the Methods, instead of just a relative intensity setting from the imaging software. Similarly,more details of the imaging system should be provided where appropriate (e.g., detector specifications).

We have now measured the intensity of the laser illumination and added this information in the

Methods: "Laser power was typically set to 0.3% to 0.8%, which yields 0.51 to 1.37 µW at 561 nm (measured with a ThorLabs Microscope Slide Power Sensor, #S170C)."

Regarding the imaging system and the detector, we provide all the information that is available to us on the microscope's technical sheets.

• The versions of analysis scripts associated with the manuscript should be uploaded to an online repository that permanently preserves the respective version.

The scripts are now available on gitbub and online repositories. The relevant links are included in the revised manuscript.